# Molecular physiology of pumiliotoxin sequestration in a poison frog

**Aurora Alvarez-Buylla**[1], **Cheyenne Y. Payne**[1], **Charles Vidoudez**[2], **Sunia A. Trauger**[2], **Lauren A. O'Connell**[1]*

1 Department of Biology, Stanford University, Stanford, California, United States of America, 2 Harvard Center for Mass Spectrometry, Harvard University, Cambridge, Massachusetts, United States of America

* loconnel@stanford.edu

**Data Availability Statement:** All LC-MS/MS data from the alkaloid analysis, the D. tinctorius transcriptome, TMM expression and count data for different tissues, fasta files with nucleotide and amino acid sequences of CYP2D6-like proteins,

## Abstract

Poison frogs bioaccumulate alkaloids for chemical defense from their arthropod diet. Although many alkaloids are accumulated without modification, some poison frog species can metabolize pumiliotoxin (PTX **251D**) into the more potent allopumiliotoxin (aPTX **267A**). Despite extensive research characterizing the chemical arsenal of poison frogs, the physiological mechanisms involved in the sequestration and metabolism of individual alkaloids remain unclear. We first performed a feeding experiment with the Dyeing poison frog (*Dendrobates tinctorius*) to ask if this species can metabolize PTX **251D** into aPTX **267A** and what gene expression changes are associated with PTX **251D** exposure in the intestines, liver, and skin. We found that *D. tinctorius* can metabolize PTX **251D** into aPTX **267A**, and that PTX **251D** exposure changed the expression level of genes involved in immune system function and small molecule metabolism and transport. To better understand the functional significance of these changes in gene expression, we then conducted a series of high-throughput screens to determine the molecular targets of PTX **251D** and identify potential proteins responsible for metabolism of PTX **251D** into aPTX **267A**. Although screens of PTX **251D** binding human voltage-gated ion channels and G-protein coupled receptors were inconclusive, we identified human CYP2D6 as a rapid metabolizer of PTX **251D** in a cytochrome P450 screen. Furthermore, a CYP2D6-like gene had increased expression in the intestines of animals fed PTX, suggesting this protein may be involved in PTX metabolism. These results show that individual alkaloids can modify gene expression across tissues, including genes involved in alkaloid metabolism. More broadly, this work suggests that specific alkaloid classes in wild diets may induce physiological changes for targeted accumulation and metabolism.

## Introduction

Animals have evolved many ways of avoiding predation to enhance their survival and fitness. One strategy is evolving chemical defenses by carrying small molecules that are toxic or repellant to predators. While some organisms are able to produce their own chemical defenses, many animals accumulate and store compounds found in their environment. Although

and code is either available as supplementary materials or are available on the Dryad Digital Repository (https://doi.org/10.5061/dryad.ns1rn8pr3). All Illumina fastq files are available on the Sequence Read Archive (https://www.ncbi.nlm.nih.gov/sra/PRJNA801674).

**Funding:** This work was supported by the National Science Foundation (NSF, www.nsf.gov, IOS-1822025) and New York Stem Cell Foundation (NYSCF-R-NI58) to LAO. AAB is supported by a NSF Graduate Research Fellowship (DGE-1656518) and an HHMI Gilliam Fellowship (www.hhmi.org, GT13330). LAO is a New York Stem Cell – Robertson Investigator (https://nyscf.org). The funders had no role in study design, data collection and analysis, decision to publish, or preparation of the manuscript.

**Competing interests:** The authors have declared that no competing interests exist.

environmentally derived compounds can be sequestered unchanged, small molecule metabolism is also used for storage or diversification of chemical repertoires. For example, European cinnabar moth (*Tyria jacobaeae*) larvae convert toxic pyrrolizidine alkaloids obtained from their plant diet into inert forms for storage throughout development [1]. In plants, the expansion of cytochrome P450 genes has led to diversification of defensive alkaloids used to combat herbivores and pathogens [2]. Uncovering general principles of how organisms sequester and metabolize chemical defenses allows for a better understanding of how animals evolve to maximize the use of their available environmental chemicals.

Poison frogs (Family Dendrobatidae) are chemically defended against predators [3–5] using alkaloids sequestered from dietary arthropods [6, 7]. Within the poison frog clade, defensive alkaloids have originated at least three times [8, 9]. Notably, chemical defense covaries with the diversification of parental care strategies in this clade [10], where in some species mothers provision chemical defenses to their tadpoles by feeding alkaloid-containing trophic eggs [11–13]. Lab feeding experiments have found that alkaloids accumulate primarily on the skin, although detectable quantities are also found in the liver and intestines [14]. This accumulation can occur within a few days [14–16], and is associated with changes in gene expression and protein abundance across tissues [14, 16]. However, previous analyses of gene expression have been limited to comparing frogs with alkaloids to those without alkaloids, leaving a major gap in our understanding of how frog physiology changes in response to specific alkaloids rather than overall toxicity. Filling this knowledge gap is important because poison frogs carry many different alkaloid classes, each of which could potentially induce specific changes in physiology.

Controlled alkaloid-feeding experiments have been crucial in understanding alkaloid metabolism in poison frogs [15, 17, 18]. Many poison frog alkaloids have been found in the ants and mites they consume, suggesting most alkaloids are sequestered unchanged [19–22]. Alkaloids of the pumiliotoxin (PTX) family are found across the poison frogs clade, including *Dendrobates*, *Oophaga*, *Epipedobates*, *Ameerega*, and *Phyllobates* species, and are thought to be sourced from formicine ants and oribatid mites [19, 20, 23, 24]. Pumiliotoxins have also been found in Malagasy poison frogs (family Mantellidae), a convergent evolution of anuran chemical defenses found in Madagascar [25]. Along with sequestering pumiliotoxins from ants, laboratory feeding experiments have also shown that some dendrobatid species metabolize PTX **251D** into allopumiliotoxin (aPTX) **267A** [15]. After PTX **251D** feeding, both PTX **251D** and its metabolite, aPTX **267A,** were detected in the skin of *Dendrobates auratus*, but only PTX **251D** was detected in the skin of *Phyllobates bicolor* and *Epipedobates tricolor* [15]. The authors then suggested that an unidentified enzyme performs the 7'-hydroxylation of PTX **251D** into aPTX **267A** in some dendrobatid species, but not all. Whether other poison frog species can metabolize PTX **251D** into aPTX **267A** and the molecular machinery responsible remains unknown.

Of the poison frog alkaloids that have been tested *in vivo* and *in vitro*, many are considered to be toxic due to their varied effects on different ion channels in the nervous system [5]. PTX **251D** is a toxic alkaloid that causes pain, hyperactivity, convulsions and death in mice and insects [15, 26, 27]. PTX **251D** likely acts as a cardiac depressor in mice, eventually leading to cardiac arrest [28, 29]. Electrophysiology work with PTX **251D** found that it inhibited both mammalian and insect voltage-gated sodium and potassium channels [30]. PTX **251D** potently inhibits insect sodium channels, which may explain its repellent effect on mosquitos [27]. Additionally, the effects of PTX **251D** on mice were reduced by previous administration of anticonvulsants phenobarbital and carbamazepine, which target calcium and sodium channels, respectively [15]. Together, these studies provide evidence that PTX **251D** is toxic by way of inhibition of mammalian and insect ion channels, however finer scale studies of

concentration-dependent responses and other molecular targets, such as G-protein coupled receptors, are needed to better understand the activity of PTX **251D**. The molecular targets of aPTX **267A** are poorly characterized, however it induces hyperactivity, convulsions, and death when injected into mice at 2 mg/kg, a lethal dose five times lower than that of PTX **251D** [15]. This finding suggests that aPTX **267A** is more toxic than its precursor molecule PTX **251D**, and that the ability to make and sequester aPTX **267A** may be an adaptive strategy to increase toxicity in some poison frog species.

This study aimed to better understand the molecular physiology of PTX **251D** metabolism in poison frogs and further characterize potential human molecular targets. We conducted an alkaloid feeding study with the Dyeing poison frog (*Dendrobates tinctorius*) to test whether this species can metabolize PTX **251D** into aPTX **267A** and to explore gene expression changes associated with PTX **251D** exposure. We predicted metabolic enzymes involved in the hydroxylation of PTX **251D** into aPTX **267A** may be upregulated in response to their metabolic target. To functionally identify PTX **251D** target proteins that either metabolize PTX or alter neuronal function, we conducted a series of high-throughput screens with human cytochrome P450s, ion channels, and G-protein coupled receptors. This body of work provides the first in depth examination of PTX-induced changes in poison frog physiology.

## Materials and methods

### Ethics statement

All animal procedures were approved by the Institutional Animal Care and Use Committee at Stanford University (protocol number #32870). Topical benzocaine was used as an anesthesia prior to euthanasia of all animals.

### Alkaloid feeding

Lab-reared (non-toxic) *Dendrobates tinctorius* were housed in terraria with live plants, a water pool, and a shelter. Ten adult females were size-matched, randomly assigned to control or experimental groups (N = 5 per group), and then housed individually. To measure the specific effects of PTX **251D** compared to a background toxicity, the control group was fed 0.01% DHQ (Sigma-Aldrich, St. Louis, USA) in a solution of 1% EtOH and the experimental group was fed a solution of 0.01% DHQ and 0.01% PTX **251D** (PepTech, Burlington, MA, USA) in a solution of 1% EtOH in water. Each frog was fed 15 μL each day for five days by pipetting the solution directly into the mouth between 10am-12pm. On the afternoon of the fifth day, frogs were euthanized by cervical transection and the dorsal skin, liver, intestines, and oocytes were dissected into Trizol (Thermo Fisher Scientific, Waltham, USA).

### RNA extraction and library preparation

RNA extraction followed the Trizol (Thermo Fisher Scientific, Waltham, MA, USA) protocol outlined in Caty *et al.* 2019 [16] and according to the manufacturer's instructions. After the first spin, the organic layer was saved for alkaloid extraction (see below). Poly-adenylated RNA was isolated using the NEXTflex PolyA Bead kit (Bioo Scientific, Austin, USA) following manufacturer's instructions. RNA quality and lack of ribosomal RNA was confirmed using an Agilent 2100 Bioanalyzer (Agilent Technologies, Santa Clara, USA). Each RNA sequencing library was prepared using the NEXTflex Rapid RNAseq kit (Bioo Scientific). Libraries were quantified with quantitative PCR (NEBnext Library quantification kit, New England Biolabs, Ipswich, USA) and a Agilent Bioanalyzer High Sensitivity DNA chip, both according to manufacturer's instructions. All libraries were pooled at equimolar amounts

and were sequenced on four lanes of an Illumina HiSeq 4000 machine to obtain 150 bp paired-end reads.

## Transcriptome assembly and differential expression analysis

We created a reference transcriptome using Trinity [31] and filtered the raw assembly by removing contigs with BLAST hits belonging to microorganisms and invertebrates in the Swiss-Prot database [32], as these represent likely parasites, prey items, or other contaminants. Overlapping contigs were clustered using cd-hit-est [33, 34] and contigs that were less than 250bp long were removed from the assembly. We mapped the paired quality-trimmed Illumina reads to the reference transcriptome using kallisto [35]. Samples were compared across treatment groups (DHQ vs DHQ+PTX) for the skin, liver, and intestines, as these tissues contained higher levels of PTX. Differences in gene expression levels were calculated using DESeq2 [36] [$P<0.05$ false discovery rate (Benjamini–Hochberg FDR), 4-fold change]. Contigs with significant expression differences were compared to the non-redundant (nr) database using BLAST with an E-value cutoff of 1e-5. Many contigs did not have a BLAST hit or aligned to hypothetical or non-vertebrate proteins. Contigs with annotations of interest were chosen based on candidates from existing literature. Boxplots were made with R package ggplot2 (R version 3.6.3) using TMM (trimmed mean of M-values) normalized expression. All scripts are detailed in S1 File.

## Alkaloid extraction and detection

To isolate alkaloids, 0.3 mL of 100% EtOH was added to 1mL of organic layer from the Trizol RNA extraction, inverted 10 times, and stored at room temperature for 2–3 minutes to precipitate genomic DNA, which was pelleted by centrifugation at 2000g for 5 minutes at 4˚C. Then, 300 μL of supernatant was transferred to a new microfuge tube. Proteins were precipitated by adding 900 μL of acetone, mixing by inversion for 10–15 seconds, incubating at room temperature for 10 min, and centrifuging at max speed for 10 min at 4˚C. Then, 1 mL of the supernatant containing alkaloids was moved into a glass vial and stored at -20˚C until dried down completely under a gentle nitrogen gas flow.

Samples were resuspended in 200 μl of methanol:chloroform 1:1 and 1 μM Nicotine-d3 (used as an internal standard). A 10-point standard curve was prepared in the same solution with DHQ and PTX **251D**. A QE+ mass spectrometer coupled to an Ultimate3000 LC (ThermoFisher) was used for analysis. Five μl of each sample were injected on a Gemini C18 column (100x2mm, Phenomenex). The mobile phases were A: water and B: acetonitrile, both with 0.1% formic acid. The gradient was 0% B for 1 min, then increased to 100% B in 14 min, followed by 5 min at 100% B and 3.5 min at 0% B. Data were quantified using accurate mass, using the standard curve for DHQ and PTX **251D** for absolute quantification. aPTX **267A** was identified by accurate mass and MS/MS fragmentation similarity to PTX.

## Alkaloid statistical analyses

R version 3.6.3 was used for all statistical analyses, and all plotting and statistics code is provided in a S1 File. There were instances in the LC-MS/MS data where the molecules of interest (DHQ, PTX **251D**, or aPTX **267A**) were not detected, and these were converted to zeros prior to statistical analyses and visualization. A generalized linear mixed model (glmmTMB package in R [37]) was used to test for differences in alkaloid abundance across tissues and treatment type with the frog as a random effect, using a negative binomial error distribution and a single zero-inflation parameter applied to all observations. PTX **251D** and DHQ were analyzed separately. The abundance of aPTX **267A** was approximated using the area-under-the-curve

divided by the internal nicotine standard, and as there is no standard for aPTX **267A,** exact pmol values could not be calculated. A Wilcoxon rank-sum test (wilcox.test) was used to compare the aPTX values in the skin between treatment groups and the Kruskal-Wallis test (kruskal.test) with a post-hoc Dunn test (dunnTest from the FSA package [38]) was used to compare the aPTX values across tissues. Boxplots used to visualize alkaloid abundance values were created in R using ggplot.

## Assays for PTX target proteins

Assays for human CYP activity (CYP phenotyping), CYP2D6 metabolite discovery (metID), ion channel inhibition (CiPA), and G-protein coupled receptor (GPCR) activity (gpcrMAX) were performed through Eurofins Discovery Services (Eurofins Panlabs Inc, St. Charles MO, USA). In the CYP phenotyping panel, human recombinant CYP1A2, CYP2B6, CYP2C8, CYP2C9, CYP2C19, CYP2D6, and CYP3A4 were included in the assay. A concentration of 1E-07 M of PTX **251D** was tested, and the percent compound remaining was quantified at 0, 15, 30, 45, and 60 minutes using HPLC-MS/MS. As a positive control, a reference compound was used for each CYP tested (see S1 File for results). In the CYP2D6 metabolism assay, human recombinant CYP2D6 was incubated with an initial concentration of 10 μM and metabolites were quantified after 90 minutes using LC-MS/MS. Two replicates were used for the CYP phenotyping panel and one replicate was used for CYP2D6 metabolite assay. The CiPA assay was a cell-based QPatch on voltage-gated sodium channel NaV1.5 (peak and late/agonist), voltage-gated potassium channels Kv4.3/KChIP2, hERG, KCNQ1/minK, Kir2.1, and voltage-gated calcium channel Cav1.2. The concentrations of PTX **251D** assayed were 3, 10, and 30 μM, and a reference compound for each channel was used (see S1 File for results and individual QPatch parameters). In the gpcrMAX assay a panel of 165 GPCRs was tested through the DiscoverX PathHunter beta-arrestin enzyme fragment complementation technology, and a concentration of 10 μM PTX **251D** was tested for both agonist and antagonist activity along with a reference compound for each GPCR (see S1 File for results).

## Identification and comparison of poison frog CYP2D6

The human CYP2D6 protein sequence (uniprot: P10635) was used to identify *D. tinctorius* homologs by sequence similarity using tblastn. We then obtained the top eight BLAST contigs from the TMM expression matrix for each tissue. A pairwise t-test was used to determine if the mean expression value was significantly different between treatments in each tissue and p-values were corrected for multiple testing using benjamini-hochberg correction. The only remaining significant difference was one contig in the gut (TRINITY_DN15846_c0_g2). Using the largest ORF protein sequence of this gene we used tblastn against BLAST nucleotide databases made from previously assembled unpublished transcriptomes or genomes in five other poison frog species: *Oophaga sylvatica* (transcriptome), *Ranitomeya imitator* (transcriptome), *Epipedobates tricolor* (transcriptome), *Allobates femoralis* (genome), and *Mantella aurantiaca* (transcriptome, Family Mantellidae, an independent origin of chemical defense sequestration in amphibians [25]). The *D. auratus* and *P. bicolor* sequences were found using the same method from previously published transcriptomes [39, 40]. The *D. tinctorius* TRINITY_DN15846_c0_g2 protein sequence and largest ORF of the top BLAST hit from each of these transcriptomes was translated and aligned to the human CYP2D6 protein sequence to identify amino acid differences in important binding residues [41]. For the *B. bufo* (XP_040266170), *R. temporaria* (XP_040185037), and *X. laevis* (XP_031756601) sequences, the nucleotide sequence of *D. tinctorius* TRINITY_DN15846_c0_g2 was used to blastn against the NR database using the BLAST web portal, and the protein sequence of the top hit from

each species was downloaded and aligned. Benchling software (Benchling Inc., San Francisco, CA) was used to find the largest ORFs, translate into amino acid sequences, and create a protein alignment using Clustal Omega with the human CYP2D6 sequence as a reference. All nucleotide sequences and amino acid sequences are included in S1 File (FASTA files).

## Results

### The Dyeing poison frog metabolizes PTX to aPTX

We conducted a feeding experiment to determine if the Dyeing poison frog (*Dendrobates tinctorius*) can metabolize PTX **251D** into aPTX **267A** (Fig 1A). Alkaloids were most abundant in the skin and liver, followed by the intestines, and only trace amounts were detected in oocytes. DHQ abundance did not differ by treatment group (GLMM treatment, p = 0.377), confirming both groups were fed equal amounts. DHQ abundance differed across tissue types (GLMM tissue, $X^2$ (3) = 203.642, p < 2e-16), with the highest levels occurring in the liver and skin (Fig 1B). PTX **251D** abundance differed by tissue and treatment (GLMM tissue*treatment, $X^2$ (3) =

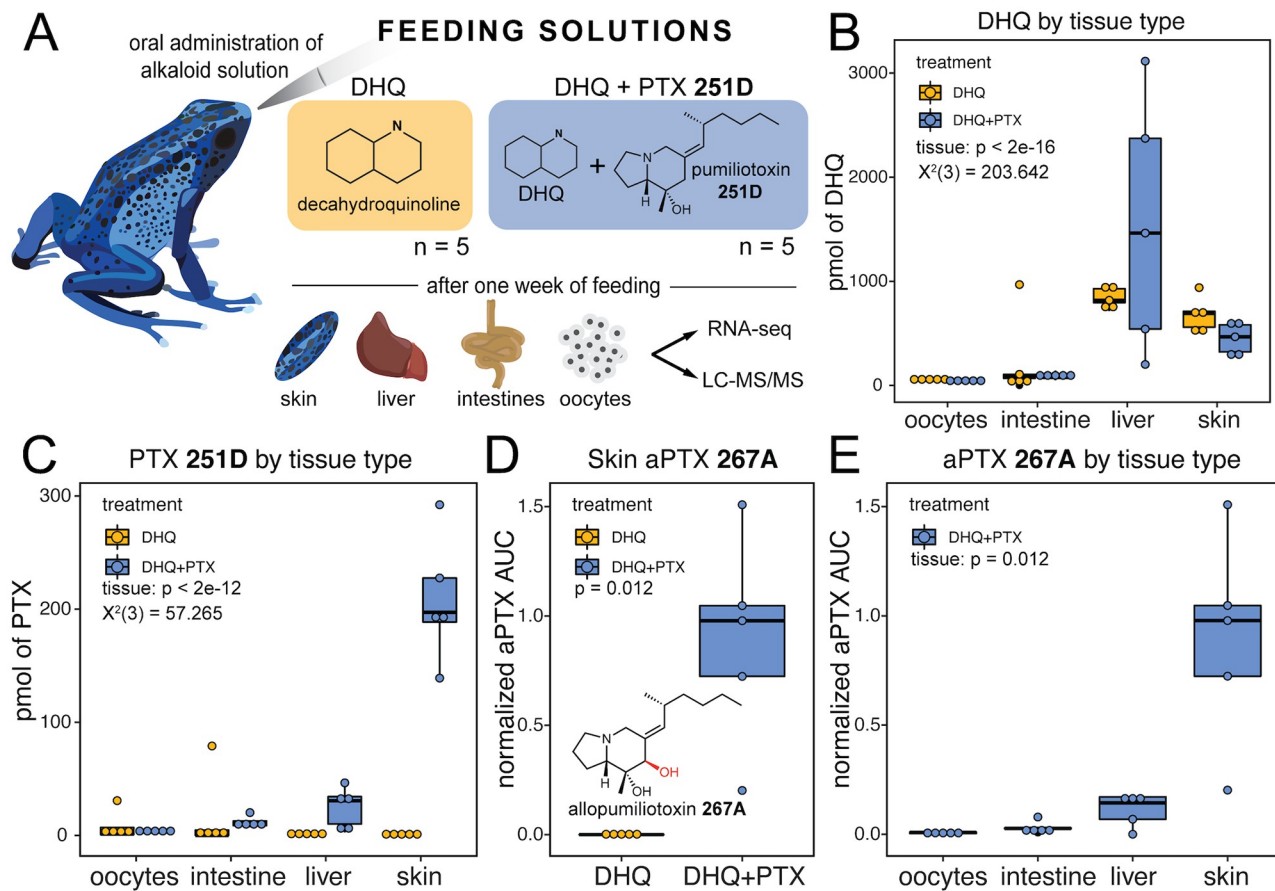

**Fig 1. Alkaloid sequestration in different tissue types.** Boxplots showing abundance of different compounds in different tissues and treatment types, with DHQ-fed frogs in yellow and DHQ+PTX-fed frogs in blue. **(A)** Frogs were orally administered either DHQ or DHQ+PTX once a day for five days. **(B)** DHQ abundance differed by tissue but not treatment group and was highest in the liver and skin (GLMM tissue, $X^2$ (3) = 203.642, p < 2e-16). **(C)** PTX levels differed by tissue and treatment, and were higher in the liver and skin of the DHQ+PTX fed group (GLMM tissue:treatment, $X^2$ (3) = 57.265, p < 2e-12). **(D)** The hydroxylated metabolite aPTX was found in the DHQ+PTX fed frogs (Wilcoxon test, W = 0, p-value = 0.012, n = 5). **(E)** aPTX abundance differed across tissues within the DHQ-PTX group (Kruskal-Wallis, $X^2$ (3) = 13.727, p = 0.003), and was found primarily in the skin, with some in the liver.

57.265, p < 2e-12), with the highest levels in the liver and skin in the DHQ+PTX feeding group (Fig 1C). We detected aPTX **267A** in the skin of all individuals in the DHQ+PTX feeding group at higher levels than the DHQ-fed group (Wilcoxon test, W = 25, p = 0.012, Fig 1D). The amount of aPTX **267A** differed across tissues (Kruskal-Wallis, $X^2$ (3) = 13.727, p = 0.003), with greater abundance in the skin than the oocytes (post-hoc Dunn test, p = 0.001) and intestines (post-hoc Dunn test, p = 0.035, Fig 1E). These data show *D. tinctorius* can metabolize PTX **251D** into aPTX **267A** and that some alkaloid metabolism may occur in the liver and intestines.

## PTX alters gene expression across tissues

To identify genes that may be involved in PTX sequestration and metabolism, we next quantified gene expression changes across tissues using RNA sequencing. The number of genes upregulated in response to PTX **251D** feeding were 282 in the intestines, 144 in the liver, and 197 in the skin (Fig 2). Although hundreds of genes were differentially expressed in each tissue, most did not have annotations or they aligned with unknown, hypothetical, or non-vertebrate proteins S1 File (Excel file). Cytochrome P450 (CYP3A29), an enzyme family well-known for their involvement in small molecule hydroxylation, was upregulated in the intestines (t = 4.7, log2FC = 5.72, p-adjusted = 0.0045; Fig 2A). In the liver, vitellogenin 2 (VTG2) was downregulated in the PTX feeding group (t = 3.8, log2FC = -7.73, p-adjusted = 0.0421, Fig 2B). MHC Class Iα was upregulated in both the liver (t = 5.7, log2FC = 3.49, p-adjusted = 0.0005) and intestines (t = 5.7, log2FC = 5.41, p-adjusted = 0.0001) in the presence of PTX **251D** (Fig 2A and 2B). In the skin, syntaxin 1A (STX1A) was upregulated (t = 4.0, log2FC = 2.58, p-adjusted = 0.0385) and a solute carrier family 2 protein (SLC2) was downregulated (t = -3.7, log2FC = 6.73, p-adjusted = 0.0496) in response to PTX **251D** (Fig 2C).

## Molecular targets of PTX 251D

A common reaction performed by cytochrome P450 (CYP) enzymes is the hydroxylation of small molecule substrates [42]. To identify candidate cytochrome P450s that may metabolize

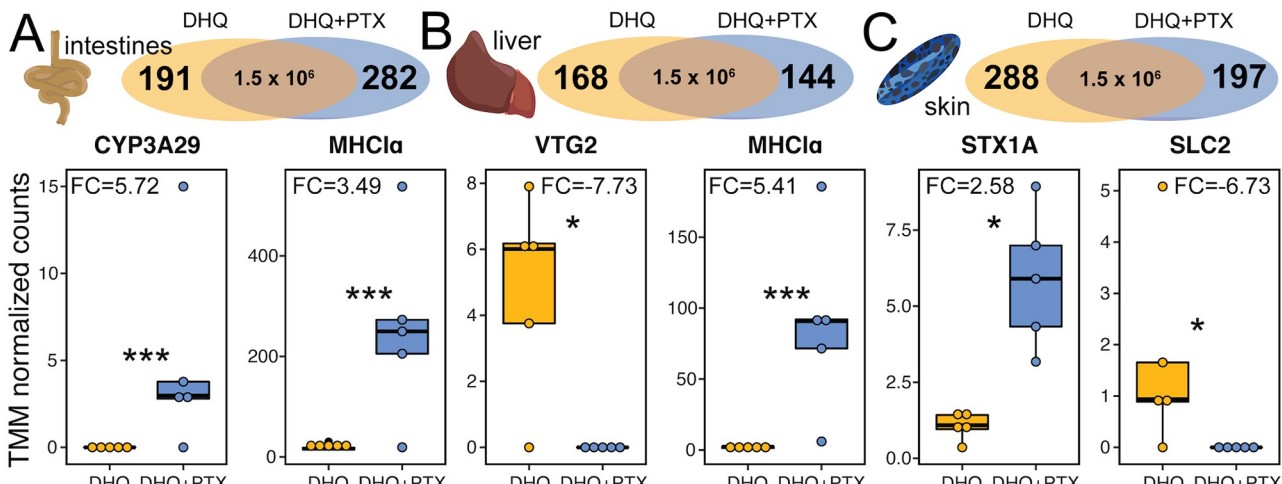

**Fig 2. Differentially expressed genes in different tissues.** Boxplots show TMM normalized expression levels in the DHQ-fed group (yellow) and DHQ+PTX fed group (blue) for a subset of differentially expressed genes. **(A)** Differentially expressed genes in the intestines include Cytochrome P450 Family 3 Protein 29 (CYP3A29) and MHC Class I alpha (MHCIα). **(B)** Differentially expressed genes in the liver include vitellogenin 2 (VTG2) and MHCIα. **(C)** Differentially expressed genes in the skin include syntaxin 1A (STX1A) and solute carrier family 2 (SLC2). (FC indicates log2 fold change values, * indicates adjusted p-value < 0.05, *** indicates adjusted p-value < 0.005; *y*-axes of individual plots have different scales).

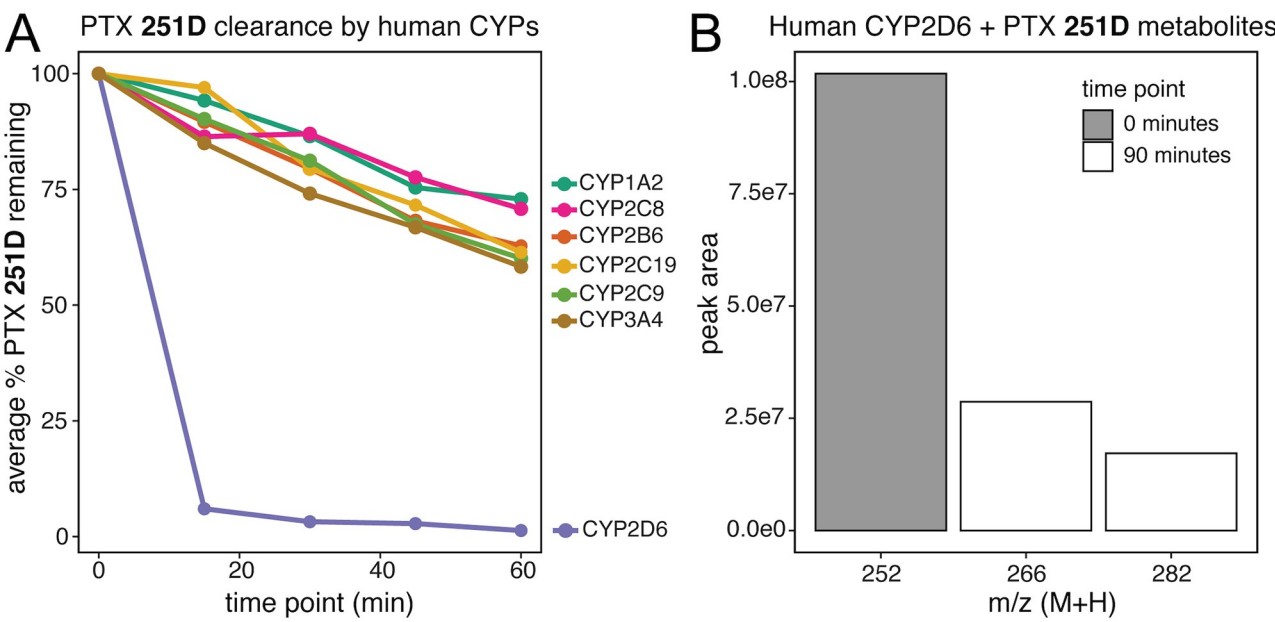

**Fig 3. Human CYP2D6 PTX 251D metabolism. (A)** Human CYP2D6 rapidly clears PTX **251D** in-vitro compared to other human CYPs. **(B)** Human CYP2D6 creates two hydroxylation products from PTX **251D** corresponding to a single hydroxylation event (m/z = 266), suggestive of aPTX **267A**, and two hydroxylation events (m/z = 282). Mass-to-charge ratio of the peak is indicated with "m/z", and "peak area" indicates the area under the curve of each peak corresponding to the m/z indicated.

PTX **251D**, we screened seven human CYPs for PTX **251D** clearance. Human CYP2D6 showed rapid clearance of PTX **251D**, with most of the initial compound depleted after 15 minutes (Fig 3A), while CYP1A2, CYP2B6, CYP2C8, CYP2C9, CYP2C19, and CYP3A4 showed minimal clearance of PTX **251D** after 60 minutes. We followed this observation with a metabolite discovery screen using human CYP2D6. PTX **251D** was completely cleared after 90 minutes, and two metabolites were identified with mass-to-charge ratio (m/z) shifts corresponding to either one (m/z of M+H = 266) or two (m/z of M+H = 282) hydroxylation events (Fig 3B). Together, this suggests that CYP2D6 may metabolize PTX **251D** into aPTX **267A**.

To better understand the physiological impacts of PTX **251D** exposure, we also conducted a screen for activity on human G-protein coupled receptors (GPCRs) and ion-channels. At a maximum concentration of 30 uM PTX **251D** showed no inhibitory effect on the ion channels NaV1.5, Kv4.3/KChIP2, hERG, KCNQ1/minK, Cav1.2, or Kir2.1. At a concentration of 10 uM PTX **251D** showed no significant inhibition or activation of any of the 168 GPCRs tested. These results suggest that at these concentrations PTX **251D** does not have an effect on the specific human GPCRs or ion channels tested in these panels.

## CYP2D6 expression and sequence diversity

Given that human CYP2D6 hydroxylates PTX **251D**, we searched for CYP2D6 homologs in the *D. tinctorius* transcriptome. Out of eight contigs with high sequence similarity to human CYP2D6, one contig showed high sequence similarity (BLAST e-value = 6.89e-121) and was upregulated in the intestines of frogs fed PTX **251D** (Welch's two-sample t-test, t = -4.0227, df = 6.3801, p-value = 0.00611; Fig 4A). We next examined if there were sequence differences in this homolog across amphibians that relate to PTX metabolism [15]. We identified homologous genes to this upregulated candidate in other amphibian transcriptomes and genomes and aligned them to the human CYP2D6 sequence to find conserved amino acid residues. Of the

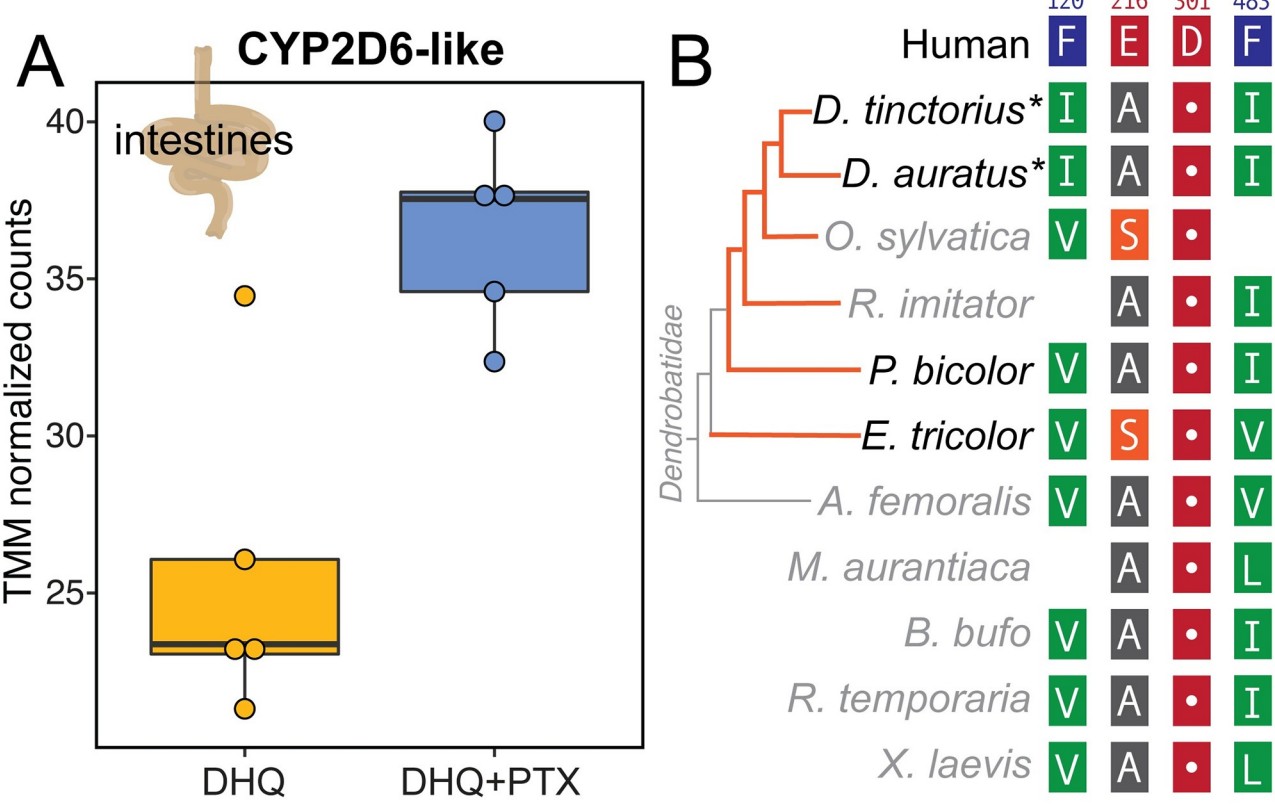

**Fig 4. CYP2D6 expression and sequence conservation. (A)** A CYP2D6-like protein was identified in the *D. tinctorius* transcriptome that was upregulated with PTX feeding. **(B)** Alignment of CYP2D6-like proteins in poison frogs and other amphibians show that important human CYP2D6 binding residue Asp301 is conserved, however other active site residues are changed in frogs. Blank spots indicate residues not present in the alignment of those species due to shorter protein sequences. The orange lines on the dendrobatid phylogeny indicate independent origins of chemical defense. Species in black have been tested for ability to metabolize PTX **251D** into aPTX **267A**, whereas species names in gray have not been tested. Asterisks (*) indicate species that sequester aPTX **267A** when fed PTX **251D**. Amino acid residues are colored using the RasMol scheme, which corresponds to amino acid properties.

human CYP2D6 main active site residues [41], Asp301 was fully conserved in all amphibian protein sequences, while Phe120 and Phe483 were changed to aliphatic residues isoleucine, valine, or leucine (Fig 4B). Glu216 in *D. tinctorius* and most other amphibians was changed to an alanine, however in *O. sylvatica* and *E. tricolor* was changed into a serine (Fig 4B). Overall, we did not identify a clear pattern in active site mutations that reflect the ability to convert PTX **251D** into aPTX **267A**, although conversion has not been tested in all species for which sequences are available.

## Discussion

Poison frogs acquire alkaloids from their diet for chemical defense, and in some cases can metabolize these compounds into more potent forms. This study shows that when fed PTX **251D,** *D. tinctorius* is able to create and sequester the more potent aPTX **267A**, and that the consumption of PTX **251D** changes gene expression across multiple tissues. Furthermore, human CYP2D6 hydroxylates PTX **251D** and a similar gene found in poison frogs and other amphibians show sequence conservation in Asp301, an important binding residue. These results expand our understanding of PTX molecular physiology, and how general alkaloid

metabolism may be co-opted for the creation of stronger chemical defense from available dietary resources.

Performing a controlled feeding study with DHQ and PTX **251D** allowed us to determine that *D. tinctorius* can metabolize PTX **251D** into aPTX **267A**. Although previous studies have documented wild *D. tinctorius* with aPTX on their skin [4, 19], this is the first experimental evidence that this species metabolizes PTX **251D** into aPTX **267A** rather than (or in addition to) sequestering aPTX from their diet. Previous work has shown that *D. auratus*, a species closely related to *D. tinctorius*, is able to convert PTX **251D** in aPTX **267A** [15]. However aPTX **267A** is not detected on the skin of *P. bicolor*, a species in the same origin of toxicity, and *E. tricolor*, a species in a different origin of aquired chemical defense in dendrobatids, when fed PTX **251D** in the laboratory [15]. It is possible that the ability to convert PTX **251D** into aPTX **267A** or sequester aPTX **267A** to the skin has only been acquired in specific poison frog populations or species. The accumulation of both DHQ and PTX **251D** in the liver, intestines, and skin, indicates that these tissues play an important role in the sequestration of alkaloids and echoes previous controlled feeding study results where frogs were only fed DHQ [14]. The liver and intestines are also important sites of alkaloid metabolism in mammals due to high levels of Cytochrome P450s [43–45]. Together, these results show that *D. tinctorius* can metabolize PTX **251D** into aPTX **267A** and that the tissue distribution of alkaloids includes the skin, liver, and intestines.

PTX **251D** feeding resulted in gene expression changes in the intestines, liver, and skin, suggesting a single alkaloid can influence poison frog physiology. Specifically, the upregulation of CYP3A29 in response to PTX in the intestines implicates this enzyme in the metabolism of PTX **251D** into aPTX **267A**, or of PTX into a metabolic byproduct to be later discarded. Although we originally expected to identify metabolism enzymes in the liver, it is possible the liver instead acts primarily as a detoxification site. In the dendrobatid *Oophaga sylvatica*, feeding DHQ compared to vehicle control leads to a downregulation of CYP3A29 in the intestines [14]. This suggests that expression of CYP3A29 may be downregulated in response alkaloids generally, yet upregulated in response to specific alkaloids, such as in the presence of PTX **251D**. The upregulation of MHC class Iα proteins in the intestines and liver in response to PTX **251D** supports previous findings that frog immune systems respond to alkaloids [16, 46]. We also found that VTG2 (vitellogenin-2) was downregulated in response to PTX **251D**. Although vitellogenins are typically thought to be egg-yolk proteins, they also play regulatory roles and protect cells from reactive oxygen species that may arise from alkaloid metabolism [47–49]. Finally, SLC2 (solute carrier family 2) which encodes for the GLUT family of glucose transporters, was downregulated in the skin with PTX feeding. Alkaloids can be potent inhibitors of GLUTs in mammalian cell lines [50], and the downregulation of GLUTs in this case may be due to the presence of concentrated PTX **251D** in the frog skin. Together, these gene expression changes in response to PTX **251D** compared to an alkaloid-fed control support an argument for physiological "fine-tuning" of gene expression in response to certain alkaloids.

Human CYP2D6 can hydroxylate PTX **251D**, and similar sequences in poison frogs and other amphibians show varying sequence conservation in important binding residues. This study provides the first evidence that human CYP2D6 rapidly clears PTX **251D**, and a gene in *D. tinctorius* with high similarity (BLAST e-value = 6.89e-121) showed an increased expression in the intestines of frogs fed PTX **251D**. Human CYP2D6 is documented to be involved in the metabolism of many compounds, including plant alkaloids [51]. Although there is no direct evidence for the upregulation of CYP2D6 in response to exogenous compounds in mammals, increased abundance of members of the CYP2D family through gene duplications in humans and mice are hypothesized to allow species to deal with the presence of dietary or environmental alkaloids [52]. Furthermore, human CYP2D6 is known to perform hydroxylation reactions

mediated through the Asp301 residue binding to nitrogen in lipophilic substrate compounds [41]. Crystal structure and site-directed mutagenesis studies have found that additional residues involved in substrate binding in the active site of human CYP2D6 are Phe120, Glu216, and Phe483 [41]. Our results find that a hydroxylation product of PTX **251D** is made by human CYP2D6, however it is not possible with our data to know if the position of hydroxylation corresponds exactly to aPTX **267A.** Nevertheless, given that Asp301 is conserved across distant frog species, many species may have the ability to create aPTX **267A,** whereas only certain poison frog species may have evolved the ability to sequester aPTX **267A** to their skin for chemical defense. *D. auratus* and *D. tinctorius* are the only tested species known to sequester aPTX **267A** when fed PTX **251D,** and the only species found with Ile120 instead of Val120 in their CYP2D6-like proteins. Although isoleucine and valine are similar amino acids this change may still influence the substrate binding affinity of CYP2D6 or positioning of substrate hydroxylation. It is also likely that there are additional residues that coordinate specificity for PTX **251D** that are yet undiscovered. Controlled feeding experiments with PTX **251D** in other species and detailed *in vitro* biochemistry and crystallography studies would need to confirm how binding specificity may differ between species that can and cannot make aPTX **267A.** Together, these data suggest that many species of amphibians may be able to hydroxylate PTX **251D**, and that *D. tinctorius* may be modulating the expression of a CYP2D6-like protein to metabolize PTX **251D** into aPTX **267A** for improved chemical defense.

We did not find any effect of PTX **251D** on the six ion channels and 168 GPCRs tested. For the ion channels, this is probably because the concentrations tested were much lower than that of previous electrophysiology work on PTX **251D** [30]. Patch clamp experiments with mammalian and insect voltage-gated sodium and potassium channels found that at 100 μM PTX **251D** inhibits both types of channels, with the strongest effect being on hKv1.3 [30]. We tested a range of concentrations of PTX **251D** in the ion channel screen based on the active concentrations tested in electrophysiology studies of batrachotoxin (BTX) [53], and the lack of effect seen by PTX **251D** at these concentrations supports findings that PTX **251D** is not as potent of a toxin as BTX for mammalian ion channels [15, 54]. To better understand the effects of different poison frog toxins, including aPTX **267A**, further systematic testing of potential targets would be required, although synthesis of the compounds found on wild poison frogs is difficult and has limited research progress in this direction.

In summary, this study provides evidence that *D. tinctorius* can metabolize PTX **251D** into aPTX **267A** and that PTX **251D** exposure changes gene expression across tissues, demonstrating that specific alkaloids can change poison frog physiology. Following up on candidate genes with biochemical studies is needed to fully characterize the genetics of alkaloid sequestration and metabolism. In the wild, where chemically defended dendrobatids carry many different alkaloids, subtle alkaloid differences may induce distinct gene expression changes. More broadly, modulating gene expression in response to specific alkaloids may set the stage for local adaptation to environmental resources.

## Supporting information

**S1 File.**
(ZIP)

## Acknowledgments

We thank Stephanie Caty and Nora Moskowitz for their comments on early versions of this manuscript. We also thank the O'Connell Lab for frog colony care and general experimental

advice and encouragement. We acknowledge that this research was conducted at Stanford University, which is located on the ancestral and unceded land of the Muwekma Ohlone tribe.

## Author Contributions

**Conceptualization:** Aurora Alvarez-Buylla, Lauren A. O'Connell.

**Data curation:** Aurora Alvarez-Buylla, Charles Vidoudez.

**Formal analysis:** Aurora Alvarez-Buylla.

**Funding acquisition:** Lauren A. O'Connell.

**Investigation:** Aurora Alvarez-Buylla, Cheyenne Y. Payne, Charles Vidoudez.

**Methodology:** Charles Vidoudez, Sunia A. Trauger.

**Project administration:** Lauren A. O'Connell.

**Resources:** Sunia A. Trauger, Lauren A. O'Connell.

**Supervision:** Sunia A. Trauger, Lauren A. O'Connell.

**Writing – original draft:** Aurora Alvarez-Buylla.

**Writing – review & editing:** Cheyenne Y. Payne, Charles Vidoudez, Sunia A. Trauger, Lauren A. O'Connell.

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
