## [Decision Letter · Decision Letter 0]

19 Jan 2022

PONE-D-21-39716Molecular physiology of pumiliotoxin sequestration in a poison frog

PLOS ONE

Dear Dr. O'Connell,

Thank you for submitting your manuscript to PLOS ONE. After careful consideration, we feel that it has merit but does not fully meet PLOS ONE’s publication criteria as it currently stands. Therefore, we invite you to submit a revised version of the manuscript that addresses the points raised during the review process.

Please address in full the minor concerns expressed by the two reviewers.

We look forward to receiving your revised manuscript.

Kind regards,

Israel Silman

Academic Editor

PLOS ONE

Journal Requirements:

"This work was supported by the National Science Foundation (NSF, www.nsf.gov, IOS-1822025) and New York Stem Cell Foundation (NYSCF-R-NI58) to LAO. AAB is supported by a NSF Graduate Research Fellowship (DGE-1656518) and an HHMI Gilliam Fellowship (www.hhmi.org, GT13330). LAO is a New York Stem Cell – Robertson Investigator (https://nyscf.org). The funders had no role in study design, data collection and analysis, decision to publish, or preparation of the manuscript."

"This work was supported by the National Science Foundation (NSF, www.nsf.gov, IOS-1822025) and New York Stem Cell Foundation (NYSCF-R-NI58) to LAO. AAB is supported by a NSF Graduate Research Fellowship (DGE-1656518) and an HHMI Gilliam Fellowship (www.hhmi.org, GT13330). LAO is a New York Stem Cell – Robertson Investigator (https://nyscf.org). The funders had no role in study design, data collection and analysis, decision to publish, or preparation of the manuscript."

4. We note that you have referenced Allobates femoralis (unpublished genome), which has currently not yet been accepted for publication. Please remove this from your References and amend this to state in the body of your manuscript: (ie “Bewick et al. [Unpublished]”) as detailed online in our guide for authors

Reviewers' comments:

Reviewer's Responses to Questions

**Comments to the Author**

1. Is the manuscript technically sound, and do the data support the conclusions?

Reviewer #1: Yes

Reviewer #2: Yes

2. Has the statistical analysis been performed appropriately and rigorously? 

Reviewer #1: Yes

Reviewer #2: Yes

3. Have the authors made all data underlying the findings in their manuscript fully available?

Reviewer #1: Yes

Reviewer #2: Yes

4. Is the manuscript presented in an intelligible fashion and written in standard English?

Reviewer #1: Yes

Reviewer #2: Yes

5. Review Comments to the Author

Reviewer #1: • What are the main claims of the paper and how significant are they for the discipline?

Though not the first paper to demonstrate PTX 251D is biotransformed into the more potent aPTX 267A in Dendrobates (the process was previously identified in D. auratus), this work provides a good launching point to better understand the molecular physiology at the level of the organism (rather than only the chemical process the compound undergoes during bioconversion, the prior extent of our knowledge), and identifies PTX 251D � aPTX 267A in a second Dendrobates species (D. tinctorius). Alvarez-Buylla et al. contributes to an exciting new area of research on how molecular responses of poison frogs may vary (both in particular genes targeted and levels of gene expression) to specific alkaloids and thereby opens the door to more granular work on how variation in environmental alkaloids may fine-tune poison frog chemical defense.

• Are the claims properly placed in the context of the previous literature? Have the authors treated the literature fairly?

Yes, and the authors conclude (fairly) that further biochemical work will be needed to elucidate the full metabolic pathways involved in alkaloid metabolism (line 386) and that each alkaloid in a complex frog chemical profile may affect gene expression levels as well as particular genes expressed (line 388), with these profiles subject to change based on environmental (dietary) factors. Thus, more experiments would be needed to tease apart the effects of specific alkaloids. The authors also recognize a limitation (that synthetic chemical studies of poison frog alkaloids is not currently a highly active research area [line 382]).

• Do the data and analyses fully support the claims? If not, what other evidence is required?

Yes.

• PLOS ONE encourages authors to publish detailed protocols and algorithms as supporting information online. Do any particular methods used in the manuscript warrant such treatment? If a protocol is already provided, for example for a randomized controlled trial, are there any important deviations from it? If so, have the authors explained adequately why the deviations occurred?

Methodologies used in this study are either established or explained in detail, so publication of protocols is not needed.

• If the paper is considered unsuitable for publication in its present form, does the study itself show sufficient potential that the authors should be encouraged to resubmit a revised version?

N/A

• Are original data deposited in appropriate repositories and accession/version numbers provided for genes, proteins, mutants, diseases, etc.?

Yes.

• Does the study conform to any relevant guidelines such as CONSORT, MIAME, QUORUM, STROBE, and the Fort Lauderdale agreement?

N/A.

• Are details of the methodology sufficient to allow the experiments to be reproduced?

Yes.

• Is any software created by the authors freely available?

N/A

• Is the manuscript well organized and written clearly enough to be accessible to non-specialists?

Yes.

• Is it your opinion that this manuscript contains an NIH-defined experiment of Dual Use concern?

No.

Additional Comments:

—

1. Correct the statement, “within the poison frog clade, the ability to sequester dietary alkaloids has evolved independently at least three times” (lines 58–59)—the two papers cited to support this statement only show that bright color and diet specialization have independently evolved but do not get at sequestration ability. Instead say something like, “defensive alkaloids have originated at least three times.”

2. It is worth also citing the Daly et al. (2003) PNAS paper (Evidence for an enantioselective pumiliotoxin 7-hydroxylase in dendrobatid poison frogs of the genus Dendrobates [see their figure 3], reference 19) for the assertion that accumulation of alkaloids takes just a few days (line 64).

3. PTXs in poison frogs are also known from mites, not only ants as you imply in line 76 (see Saporito et al. [2007] in PNAS, Oribatid mites as a major dietary source for alkaloids in poison frogs, Saporito et al. [2011] in J Chem Ecol, Alkaloids in the mite Scheloribates laevigatus: further alkaloids common to oribatid mites and poison frogs, and Takada et al. [2005] in J Chem Ecol, Scheloribatid mites as the source of pumiliotoxins in dendrobatid frogs).

4. The citation given in line 82 (reference 10) is almost certainly incorrect. Should be Daly et al. (2003) (reference 19).

5. Line 100 again needs to cite reference 19 and make it clear that the experimental results on aPTX 267A toxicity are also from this study.

Reviewer #2: I enjoyed reading this excellent manuscript. This is an elegant study targeting a very relevant question in the evolution of amphibian toxicity, and presents a series of gene expression and protein assay experiments that provide important new insights into the mechanism that underlies biomodification of alkaloids by dendrobatid frogs.

The manuscript is excellently written, with basically no typo, and my review therefore is much shorter than usual, as I have almost nothing to criticize. The following are just a few minor points that the author should consider in the revision of the manuscript. I look forward to seeing this great study published!

Line 176 and elsewhere: the tissue type designation "eggs" is misleading; I very much suppose what is meant here is oocytes and not mature eggs. Please change terminology throughout, including in the figures.

Line 76: Oribatid mites are at least a second, if not the major source of pumiliotoxins besides ants; see Takada et al. doi: 10.1007/s10886-005-7109-9 as well as Saporito et al. https://doi.org/10.1073/pnas.0702851104 and other papers.

The list of references requires attention. I guess it was made with a reference managing system which of course is fine but has the drawback of formatting errors. For instance, scientific names are not italicized, and in numerous references, all words are in capitals while they should not be (I noticed the latter at least in refs # 17, 24, 38, 50 but I may have overlooked some).

SRA accession numbers need to be added before final publication.

6. PLOS authors have the option to publish the peer review history of their article (what does this mean?). If published, this will include your full peer review and any attached files.

Reviewer #1: No

Reviewer #2: No

---

## [Author Response · Author response to Decision Letter 0]

8 Feb 2022

We have responded to each reviewer comment (see the response to reviewers below) and we have also made the following changes requested by the editor: 

1. Ensured that all files in the submission meet all of the PLOS ONE style requirements.

2. Reviewed the reference list to make sure that all citations are correct.

3. Removed the funding information from the “Acknowledgement” section of the text of the manuscript. The funding statement as originally submitted through the portal is correct and does not require amendment.

4. Removed the “(unpublished genome)” as a reference in the methods.

5. Included a section in the Methods named “Ethics Statement” which includes the animal ethics board information, protocol number, and anesthesia used prior to euthanasia. 

Response to reviewers:

We thank the reviewers for their thoughtful comments on our manuscript. We have addressed each of the comments below. 

Reviewer 1:

1. Correct the statement, “within the poison frog clade, the ability to sequester dietary alkaloids has evolved independently at least three times” (lines 58–59)—the two papers cited to support this statement only show that bright color and diet specialization have independently evolved but do not get at sequestration ability. Instead say something like, “defensive alkaloids have originated at least three times.”

Response: We agree that the above statement does not reflect the papers cited, and have corrected it to read “Within the poison frog clade, defensive alkaloids have originated at least three times.”

2. It is worth also citing the Daly et al. (2003) PNAS paper (Evidence for an enantioselective pumiliotoxin 7-hydroxylase in dendrobatid poison frogs of the genus Dendrobates [see their figure 3], reference 19) for the assertion that accumulation of alkaloids takes just a few days (line 64).

Response: We agree and have added this citation in line 60 (previously line 64).

3. PTXs in poison frogs are also known from mites, not only ants as you imply in line 76 (see Saporito et al. [2007] in PNAS, Oribatid mites as a major dietary source for alkaloids in poison frogs, Saporito et al. [2011] in J Chem Ecol, Alkaloids in the mite Scheloribates laevigatus: further alkaloids common to oribatid mites and poison frogs, and Takada et al. [2005] in J Chem Ecol, Scheloribatid mites as the source of pumiliotoxins in dendrobatid frogs).

Response: We apologize for missing these important references and have included them in line 70. We also changed the text of lines 68-70 to “Alkaloids of the pumiliotoxin (PTX) family are found across the poison frogs clade, including Dendrobates, Oophaga, Epipedobates, Ameerega, and Phyllobates species, and are thought to be sourced from formicine ants and oribatid mites.” 

4. The citation given in line 82 (reference 10) is almost certainly incorrect. Should be Daly et al. (2003) (reference 19).

Response: Thank you for catching this error, it has been corrected.

5. Line 100 again needs to cite reference 19 and make it clear that the experimental results on aPTX 267A toxicity are also from this study.

Response: This citation has been added as well.

Reviewer 2:

I enjoyed reading this excellent manuscript. This is an elegant study targeting a very relevant question in the evolution of amphibian toxicity, and presents a series of gene expression and protein assay experiments that provide important new insights into the mechanism that underlies biomodification of alkaloids by dendrobatid frogs.

The manuscript is excellently written, with basically no typo, and my review therefore is much shorter than usual, as I have almost nothing to criticize. The following are just a few minor points that the author should consider in the revision of the manuscript. I look forward to seeing this great study published!

Response: Thank you for your positive evaluation of our manuscript! We agree that this study provides a step forward in understanding alkaloid metabolism in poison frogs and hope that it will lead to further exploration in this area. 

Line 176 and elsewhere: the tissue type designation "eggs" is misleading; I very much suppose what is meant here is oocytes and not mature eggs. Please change terminology throughout, including in the figures.

Response: Thank you for this suggestion, we agree that the term oocytes is more appropriate given that they were not mature eggs. This has been corrected throughout the manuscript.

Line 76: Oribatid mites are at least a second, if not the major source of pumiliotoxins besides ants; see Takada et al. doi: 10.1007/s10886-005-7109-9 as well as Saporito et al. https://doi.org/10.1073/pnas.0702851104 and other papers.

Response: We apologize for missing these important references and have included them in line 70. We also changed the text of lines 68-70 to “Alkaloids of the pumiliotoxin (PTX) family are found across the poison frogs clade, including Dendrobates, Oophaga, Epipedobates, Ameerega, and Phyllobates species, and are thought to be sourced from formicine ants and oribatid mites.” 

The list of references requires attention. I guess it was made with a reference managing system which of course is fine but has the drawback of formatting errors. For instance, scientific names are not italicized, and in numerous references, all words are in capitals while they should not be (I noticed the latter at least in refs # 17, 24, 38, 50 but I may have overlooked some).

Response: Thank you for catching these errors, they have been corrected. 

SRA accession numbers need to be added before final publication.

Response: These have been included, along with a data dryad accession for the raw LC-MS/MS data. The text on lines 430-434 now reads “All LC-MS/MS data from the alkaloid analysis, the D. tinctorius transcriptome, TMM expression and count data for different tissues, fasta files with nucleotide and amino acid sequences of CYP2D6-like proteins, and code is either available as supplementary materials or are available on the Dryad Digital Repository (https://doi.org/10.5061/dryad.ns1rn8pr3). All Illumina fastq files are available on the Sequence Read Archive (https://www.ncbi.nlm.nih.gov/sra, SUB11006206).”

---

## [Editor Report · Decision Letter 1]

14 Feb 2022

Molecular physiology of pumiliotoxin sequestration in a poison frog

PONE-D-21-39716R1

Dear Dr. O'Connell,

We’re pleased to inform you that your manuscript has been judged scientifically suitable for publication and will be formally accepted for publication once it meets all outstanding technical requirements.

Kind regards,

Israel Silman

Academic Editor

PLOS ONE
---

## [Editor Report · Acceptance letter]

3 Mar 2022

PONE-D-21-39716R1 

Molecular physiology of pumiliotoxin sequestration in a poison frog 

Dear Dr. O'Connell:

I'm pleased to inform you that your manuscript has been deemed suitable for publication in PLOS ONE. Congratulations! Your manuscript is now with our production department. 

Kind regards, 

on behalf of

Prof. Israel Silman 

Academic Editor

PLOS ONE